# Evaluation of short-term hair follicle storage conditions for maintenance of RNA integrity

**Eilís E. Harkin, John A. Browne, Barbara A. Murphy** *

School of Agriculture and Food Science, University College Dublin, Belfield, Dublin, Ireland

* barbara.murphy@ucd.ie

**Data Availability Statement:** All the data relevant to the experiments described in this manuscript are provided within the manuscript as Tables.

**Funding:** Funding was received from the Morris Animal Foundation (Colorado, USA). The author

## Abstract

Hair follicles provide an easily accessible tissue for interrogating gene expression for multiple purposes in mammals. RNAlater® is a liquid storage solution that stabilises and preserves cellular RNA, eliminating the need to immediately process or freeze tissue specimens. The manufacturer advises storage of samples at 2-8˚C overnight before transfer to –20˚C. This study aimed to evaluate RNA integrity in hair follicle samples collected from horses, stabilized in RNAlater®, and stored under three short-term storage conditions. Mane hair samples complete with follicles were collected from four horses at a single time point. Approximately 15 hairs were placed in each of three 2 mL tubes containing 0.75ml RNAlater® solution. Test group A was stored at 4˚C for 24-h, then decanted and stored at -20˚C. Test groups B and C were stored at 4˚C and 19˚C (room temperature) respectively for 7 days, then decanted and stored at -20˚C. RNA was isolated from all samples and RNA quantity and quality were measured. One-way ANOVA revealed no difference in RNA concentration (A:516 +/-125 ng/ml, B:273+/-93 ng/ml, C:476+/-176 ng/ml;P = 0.2) or quality (A:9.5 +/-0.19, B:9.8+/-0.09, C:9.2+/-0.35 RIN; P = 0.46) between the test groups. There were no group differences in mean Cycle Threshold values from qPCR validation assays confirming high-quality template cDNA. The results suggest that storage of hair follicles for one week in RNAlater® at cool or room temperature conditions will not compromise RNA integrity and will permit extended transport times from remote sampling locations without the need for freezing.

## Introduction

The integrity of RNA molecules is critical for research that attempts to capture a snapshot of gene expression at the time of sample collection [1]. Poor sample handling, extended storage and inappropriate storage during the transportation of samples can all cause RNA degradation [2]. It has been proposed that poor RNA quality contributes to inaccurate results in gene expression analysis studies [3]. Good RNA quality is regarded as one of the most important factors in obtaining relevant reliable gene expression data in quantitative (q) PCR investigations [4]. The quality of RNA is assessed by determining the RNA integrity value (RIN), which ranges from 1 (indicating degraded molecules) to 10 (indicating intact molecules) based on

BAM is a collaborator on grant number D22EQ-514 awarded to Hartpury University (Gloucestershire, UK). The URL of the funder's website is https://www.morrisanimalfoundation.org/. The funders did not play any role in the study design, data collection and analysis, decision to publish or preparation of the manuscript.

**Competing interests:** The authors have declared that no competing interests exist.

evaluating the intactness RNA using RNA electrophoretic measurements and machine learning based algorithm [1].

RNAlater® is an aqueous tissue storage reagent used to stabilize the RNA content of samples and prevent RNA degradation for clinical genomic and transcriptomic research [5].The manufacturer's product technical information advises that samples should be stored in RNAlater® at 2-8˚C overnight before transfer to a -20˚C freezer, and that storage at 37˚C results in partial RNA degradation after 3 days [6]. However, animal field research is frequently conducted in remote locations or at facilities without access to freezers before or during transportation to a laboratory.

To preserve gene expression at the time of sampling and provide reliable and reproducible gene expression data by qPCR, the integrity of the RNA molecule must be preserved during sample storage with RNAlater® [7]. Live animal tissue collection methods should take into account the efficiency of sample collection, the safety of the researcher, animal welfare, and permit a practical means of sample storage and transport in order to preserve sample quality [8, 9]. All of the above are important factors for consideration in animal research, particularly where sampling occurs at locations distant from a laboratory.

Hair follicles (HF) represent a useful tissue in animal research, as sample collection is less invasive and causes minimal disturbance to the animal, offering a more practical alternative to blood collection that eliminates the need for sedation. Thus, HF can provide sufficient biological material while replacing more stressful and uncomfortable sample collection methods for animals [10, 11]. Hair follicles are also very useful in mammalian research involving recurrent sampling, to avoid repeated invasive biopsies [12]. Hair follicles have been utilised for analysing gene expression in humans, rodents, horses and wildlife populations such as the American pika *(Ochotonaprinceps)* [10–16]. To date, there is limited information on acceptable short-term storage conditions for HF in RNAlater for maintenance of RNA integrity [17, 18]. A previous study of extended human HF storage in RNAlater at different temperatures determined that RNA could be extracted from small numbers of hair follicles, but the integrity of the RNA samples was not evaluated [17].

The aim of this experiment was to evaluate RNA integrity and quantity following equine HF collection, stabilization in RNAlater® and storage at different temperatures for one week prior to RNA isolation.

## Materials and methods

This study was conducted in accordance with the 'Code of Good Practice in Research' (University College Dublin, Ireland) and 'The Directive 2010/63/EU of the European Parliament and of the Council on the Protection of Animals used for Scientific Purposes'. The study described qualified for exemption from full ethical review by University College Dublin's Animal Research Ethics Approval Committee as it met the following criterion for exemption: 'The study does not involve euthanising a living animal or conducting a *procedure* on a living animal', where '*procedure*' is defined as 'any use, invasive or non-invasive, of an animal for experimental or other scientific purposes, with known or unknown outcome, or educational purposes, which may cause the animal a level of pain, suffering, distress or lasting harm equivalent to, or higher than, that caused by the introduction of a needle in accordance with good veterinary practice' in S.I. No. 543/2012—European Union (Protection of Animals used for Scientific Purposes) Regulations 2012.

Four horses of mixed light horse breed, ranging in age from 12–16 years and maintained at University College Dublin Lyons Farm, were used for sample collection. Mane hair samples were collected from four horses at a single time point in October. Three samples were collected

from each horse as follows: 20–30 hairs were isolated, held firmly and removed using a quick downward force. Hairs were held to the light to confirm follicle attachment and then inserted into a 2mL tube containing 0.75ml RNAlater® solution, such that the follicles were submerged. The excess hair was trimmed before closing the cap and the tubes stored upright. This process was repeated if more hair follicles were required. One sample from each horse was randomly allocated to each of three test groups. Samples in test group A were stored at 4°C for 24 h, decanted and then stored at -20°C following the manufacturer's recommended protocol. Test group B was stored at 4°C for 24 h and maintained at 4°C for 7 days. Test group C was stored at 19°C (room temperature) for 24 h, and maintained at 19°C for 7 days. After 7 days, samples in test groups B and C were decanted and stored at -20°C. Removal of the RNAlater solution helps to prevent crystallization upon freezing and potential damage to the RNA molecule.

## Hair follicle preparation prior to RNA isolation

The sample tubes were removed from the freezer and placed in a -20°C storage block. Each hair sample was laid out on aluminium foil and a forceps and scissors were used to carefully trim each hair to just below the hair follicle (Fig 1). The trimmed follicles were carefully transferred to a nuclease-free 1.5 ml microcentrifuge tube using forceps. The forceps and scissors were wiped down with RNaseZap (Invitrogen, Waltham, MA, USA) between each sample and a new piece of aluminium foil was used to avoid contamination.

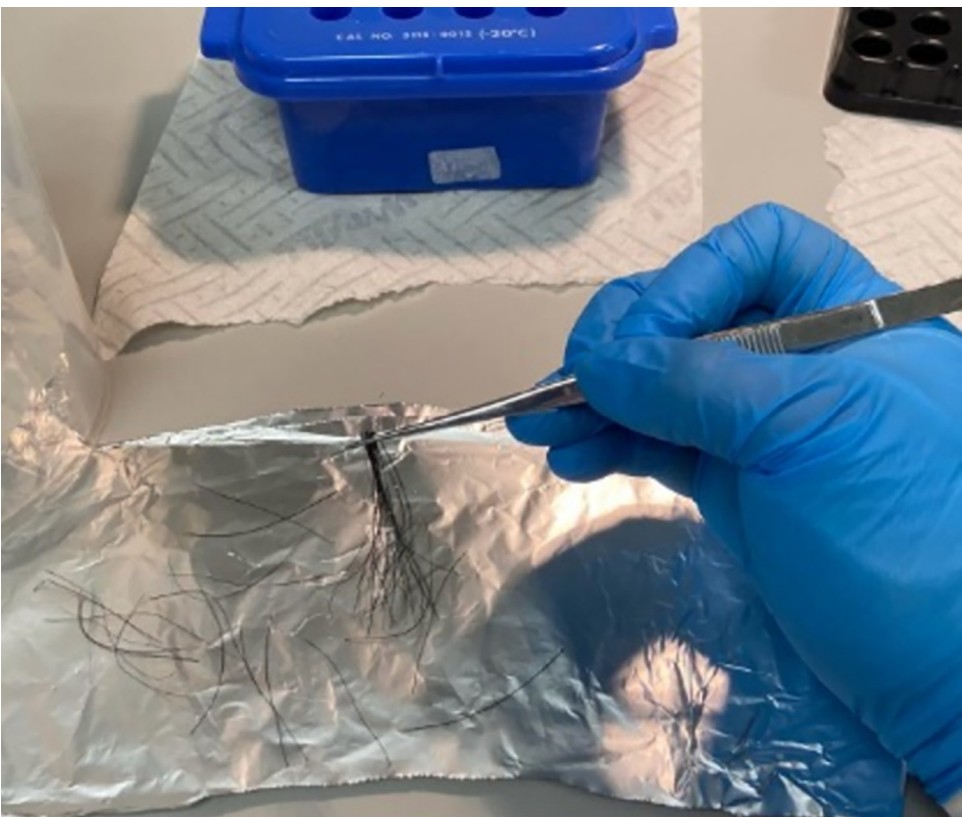

**Fig 1. Preparing to trim the hairs above the follicle.**

## RNA isolation

RNA was isolated using the *Quick-RNA™ Microprep Kit* (Zymo research, California, USA) as per the manufacturer's instructions, with some modifications: A single 5mm Qiagen steel bead was added to each tube prior to the addition of RNA Lysis Buffer. Following the addition of the Lysis Buffer the samples were homogenized for 2 minutes at a frequency of 30.0 hertz using the Qiagen TissueLyser system. Following the addition of 400 μL of RNA Wash buffer an additional spin step was carried out for 2 minutes to ensure the removal of any residual wash buffer. The RNA was eluted in 15.0 μL. One uL of the elution solution was used to measure RNA concentration using the NanoDrop Spectrophotometer (Thermo Fisher Scientific, MA, USA). RNA quality was assessed by measuring the RIN values using the Agilent Bioanalyser RNA 6000 (Agilent Technologies, CA, USA).

## cDNA conversion

For each sample, 250 ng of RNA was converted to complementary DNA (cDNA) in a 20 μL reaction using the Applied Biosystems High-Capacity cDNA Reverse Transcription Kit (Thermo Fisher) as per the manufacturer's instructions. A cDNA pool containing 3.0 μL from each sample was prepared and used to generate a 7-point, 1 in 4 serial dilution for the standard curve, positive controls and interpolate calibrators.

## Quantitative polymerase chain reaction (qPCR)

Quantitative PCR assays to detect transcript expression for ribosomal protein L19 (*RPL19*) and H3.3 histone A (*H3F3A*) using the Applied Biosystems 7500 FAST Sequence Detection system (Thermo Fisher Scientific, MA, USA) and SYBR green chemistry (Bioline, London, UK), were conducted to confirm and validate template cDNA quality. *H3F3A* and *RPL19* were selected as previous studies demonstrated their stability as reference genes in animal studies [19]. Primer design was performed using the PrimerBLAST tool (https://www.ncbi.nlm.nih.gov/tools/primer-blast/) (Table 1) and the efficiencies of the primers were determined using the standard curve method, and were shown to lie between 90% and 110%.

The PCR reactions were prepared in duplicate in a 20 μL reaction as per the manufacturer's instructions (FastStart Universal SYBR Green Master, Roche) using a final concentration of 300 nM for each primer. The thermal cycling parameters involved one cycle at 50˚C for 2 minutes, and 95˚C for 10 minutes followed by 40 cycles at 95˚C for 15 seconds, ending with a 60˚C cycle for 1 minute.

## Statistical analysis

One-way repeated measures ANOVA was used to assess group differences in RNA quantities, RIN values and qPCR Cycle Threshold (CT) values using Prism 9 version 9.1 for Mac OS. Data are presented as means +/- SEM. A P value of <0.05 was considered significant.

**Table 1. Equine primer sequences used for real-time qRT-PCR.**

| Gene Symbol | Forward 5'– 3' | Reverse 5'– 3' |
|---|---|---|
| H3F3A | CAAACTTCCCTTCCAGCGTC | TGGATAGCACACAGGTTGGT |
| RPLI9 | CTGATCATCCGGAAGCCTGT | GGCAGTACCCTTTCGCTTAC |

**Table 2. RNA quantity (ng/uL), RNA quality (RIN) and qPCR Cycle Threshold (CT) values for samples stored in RNAlater at -20˚C (Group A), 4˚C (Group B) and 19˚C (Group C) for one week prior to RNA isolation.**

| Sample Number | Horse Number | Sample storage temperature | RNA (ng/uL) | RIN | CT value H3F3A | CT value RPL19 |
|---|---|---|---|---|---|---|
| 1 | 1 | -20˚C | 446.23 | 9.4 | 20.55 | 19.10 |
| 2 | 2 | -20˚C | 608.88 | 9 | 22.61 | 19.41 |
| 3 | 3 | -20˚C | 798.87 | 9.9 | 21.18 | 20.22 |
| 4 | 4 | -20˚C | 211.31 | 9.6 | 21.42 | 20.53 |
| 5 | 1 | 4˚C | 432.85 | 10 | 20.67 | 19.71 |
| 6 | 2 | 4˚C | 150.42 | 9.8 | 20.31 | 19.94 |
| 7 | 3 | 4˚C | 430.10 | 9.7 | 20.68 | 19.17 |
| 8 | 4 | 4˚C | 81.78 | 9.4 | 20.85 | 19.59 |
| 9 | 1 | 19˚C | 248.82 | 9.5 | 21.57 | 19.77 |
| 10 | 2 | 19˚C | 448.19 | 9.7 | 20.52 | 20.38 |
| 11 | 3 | 19˚C | 979.98 | 8.2 | 20.99 | 19.49 |
| 12 | 4 | 19˚C | 226.17 | 9.6 | 20.26 | 19.61 |

## Results

One-way repeated measures ANOVA revealed no difference in RNA concentrations between test groups (A:516 +/-125ng/ml, B:273+/-93ng/ml, C:476+/-176ng/ml; P = 0.2). RIN values for all samples ranged from 8.2–10 demonstrating that the RNA was of high quality and integrity. One-way ANOVA revealed no difference in RNA quality (A:9.5 +/-0.19, B:9.8+/-0.09, C:9.2 +/-0.35; P = 0.46) between the test groups. One-way ANOVA revealed no differences between qPCR CT values between test groups for H3F3A (A:21.44+/- 0.86, B:20.63+/-0.22, C:20.26 +/-0.57; P = 0.34) or RPL19 (A:19.82+/-0.67, B:19.17+/-0.32, C:19.5+/-0.39; P = 0.7). All data are presented in Table 2.

## Discussion

This study provides the first evaluation of equine hair follicle RNA integrity following short-term storage in RNAlater® at different ambient temperatures. The results demonstrate that there is no negative impact on RNA integrity from hair follicle samples stored at room temperature or cool conditions for a period of one week prior to freezing for long-term storage. The results indicate that RNA integrity is maintained in the stabilisation solution allowing greater flexibility of storage when facilities for freezing samples during field research or transportation are unavailable.

Hair follicles offer a non-invasive and viable source of biological material for research and disease analysis as they contain cells that are a source of RNA required for gene expression profiling [18]. Animal experimentation and health evaluations are often conducted at locations distant from a laboratory with facilities for refrigeration or freezing, and where sample transportation times are lengthy. A study by Neary et al., (2014) showed that hair follicle collection from yaks (*Bos Grunniens)* could be conducted by an inexperienced handler, and was less invasive and less challenging to transport and store samples compared to venous blood sampling [20]. Hair follicles have also been utilised for the molecular identification of carcinomas [21] and traumatic brain injuries in rodents [22].

Bradley et al., (2005) were among the first to test the stability of RNA from hair follicles at various temperatures and time periods [17]. Similar to this study, HF were stored under 3 different conditions; at room temperature or -20˚C for 1, 3, 6 or 12 weeks, or room temperature

for 24 h followed by -20°C [17]. Samples were not reported to have been decanted before freezing and an additional buffer was added to the storage solution prior to processing [17]. Assessment of RNA integrity was determined by PCR amplification success of cDNA and assessed visually following separation on 1% agarose gels. This method does not allow for accurate quantification and visibility of gene-specific amplification products on the gels could only estimate concentrations in the range of 0 to >10ng/uL. The low number of follicles per sample (3 or 10 hairs) collected in the study, as well as other differences highlighted in the methodology, may potentially explain some of the failed amplifications observed by the authors for samples stored for 6 and 12 weeks. The current study improves upon the assessment of RNA integrity, albeit over a shorter storage timeframe, which is crucial for accurate quantitative gene expression results [23].

A further study seeking to determine whether intact RNA could be extracted from a small number of plucked human hair follicles reported low ribosomal RNA ratios and found that RIN values were indeterminable in most samples [18]. However, a different RNA extraction protocol to the current study was employed and initial ambient storage temperatures were not reported [18]. While lower numbers of hair follicles were used per sample, it is unclear whether this, the storage conditions or the extraction protocol was the reason for the poorer quality of RNA reported.

Many such studies aim to evaluate gene expression to better understand hair growth in order to provide potential targets for the treatment of human hair loss and other skin conditions [24, 25]. An equine study by Naboulsi et al., (2022), found that keratin-related genes were more plentiful in hair follicle samples than in skin biopsies, making HF an excellent tissue for functional studies on colouration, shape, and growth of hair as these characteristics can be attenuated in skin biopsies [26].

In the current study, lower concentrations of RNA were obtained in samples from Horse 4. Hairs from this horse appeared visibly thicker than the other sampled horses and it is likely that the larger diameter hair shafts may have increased the viscosity of the homogenate and compromised the performance of the RNA isolation kit extraction columns, reducing the concentration of RNA isolated. This highlights the importance of trimming the hairs to just below the follicles and taking care to reduce the number of hairs collected per sample in animals with particularly thick hairs.

As the horses in this study were all exposed to natural changes in environmental photoperiod throughout the year, it is assumed that their hair coats were in the anagen phase of the hair growth cycle at the sampling time in October, in response to the shortening daylight hours that stimulates active growth of the heavier winter coat [27]. As specific breed details for the study horses were unavailable, it is possible that Horse 4 was a more rustic breed. Breed rusticity is suggested to impact the response of the coat to environmental lighting changes such that at the time of sampling, this horse may have had a heavier winter coat and thicker hairs. Despite this observation, the lowest RNA quantity obtained was >1.2 μg when accounting for the elution volume, which is surplus to the requirements of any gene expression assay. These results should help refine methodologies for future animal studies to improve the reliability, repeatability and quantities of RNA achieved.

The small sample size used in this study may have represented a limitation. However, in a relevant previously published study evaluating storage conditions of hair follicles in humans, samples from only two subjects were used and only two replicates per experimental treatments were assessed [17]. Nonetheless, three biological replicates are generally advised in gene expression experiments [28] and a more recent human hair follicle transcriptome profiling study used four subjects [13]. It was therefore decided that samples from four animals maintained in identical conditions and collected at a single time point would be sufficient. Previous

published experiments conducted by the research team suggested there would be little variability expected in the biological material collected from each horse [14, 16] such that four represented a suitable sample size for assessment of the sample storage conditions where we were interested only in detecting large differences in relation to RIN values. The minimum acceptable RIN value for an RNAseq analysis from samples derived from human hair was 7 [13]. All but one sample in the current study resulted in RNA with a RIN value less than 9.4.

It would be interesting to evaluate the stability of RNA from samples collected and stored in RNAlater® solution under higher short-term storage temperatures to emulate sample collections from animals in hotter climates. Future studies could be carried out by storing the samples in incubators set at different temperatures and comparing the RNA quality and quantity obtained to samples maintained at the recommended conditions.

Several studies have utilised RNA isolated from HF to interrogate the mammalian circadian clock, an animal's internal endogenous 24-hour rhythm generator, and the relative expression of specific clock genes has been used to identify the phase of the biological clock and the strength of its synchronisation with the environment [14, 16, 29, 30]. Hair follicles represent a peripheral clock [30] that is synchronised by the master mammalian clock which resides within the suprachiasmatic nucleus (SCN) of the hypothalamus [31, 32]. Peripheral clocks are used as important phase markers of the circadian system's internal synchrony with the external environment, as the central clock in the SCN is often not easily or humanely accessible [33, 34].

Hair follicles therefore offer an efficient and non-invasive way to assess and monitor a variety of illnesses and ailments associated with circadian misalignment such as mood disorders, cancer, diabetes, cardiovascular disease and obesity [30, 35]. In domesticated and captive animals, numerous species-specific biological requirements for animal well-being may be difficult to assess [33]. In such unnatural environments, evaluating clock gene transcription in HF could be used as a biomarker for circadian irregularities caused by ill health and/or inappropriate environmental management, both of which represent animal welfare concerns [36]. Recently, clock gene expression in HF from racehorses highlighted the impact of stable lighting on circadian rhythmicity in this peripheral clock [16].

## Conclusion

We demonstrate that HF can be stored in RNAlater® in a fridge or at room temperature for one week and still provide RNA of excellent quality and quantity for quantitative gene expression analysis. Utilising a straightforward sample collection process, HF provide a convenient tissue for a wide range of functional genomics assays utilising RNA in samples from animals located in remote areas, or for monitoring circadian health and wellness for certain animal groups.

## Author Contributions

**Conceptualization:** Barbara A. Murphy.

**Formal analysis:** Eilís E. Harkin, Barbara A. Murphy.

**Investigation:** Eilís E. Harkin.

**Methodology:** Eilís E. Harkin, John A. Browne.

**Supervision:** John A. Browne, Barbara A. Murphy.

**Writing – original draft:** Eilís E. Harkin.

**Writing – review & editing:** Eilís E. Harkin, John A. Browne, Barbara A. Murphy.

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
