## [Decision Letter · Decision Letter 0]

4 Dec 2023

PONE-D-23-34914Evaluation of short-term hair follicle storage conditions for maintenance of RNA integrity.PLOS ONE

Dear Dr. Murphy,

Thank you for submitting your manuscript to PLOS ONE. After careful consideration, we feel that it has merit but does not fully meet PLOS ONE’s publication criteria as it currently stands. Therefore, we invite you to submit a revised version of the manuscript that addresses the points raised during the review process.

We look forward to receiving your revised manuscript.

Kind regards,

Abeer El Wakil, PhD

Academic Editor

PLOS ONE

Journal Requirements:

3. We note you have included a table to which you do not refer in the text of your manuscript. Please ensure that you refer to Table 1 in your text; if accepted, production will need this reference to link the reader to the Table.

Additional Editor Comments:

The concept of the present study seems interesting. In this work, the authors address an original topic and highlight valid challenges associated with sample collection in remote locations where researchers may not always be in a position to follow optimal transport and storage guidelines. However, the reviewers raised some concerns and I highly recommend the authors to address them precisely before I could deliver a decision on this work.

Reviewers' comments:

Reviewer's Responses to Questions

**Comments to the Author**

1. Is the manuscript technically sound, and do the data support the conclusions?

Reviewer #1: Yes

Reviewer #2: Partly

2. Has the statistical analysis been performed appropriately and rigorously? 

Reviewer #1: Yes

Reviewer #2: No

3. Have the authors made all data underlying the findings in their manuscript fully available?

Reviewer #1: Yes

Reviewer #2: No

4. Is the manuscript presented in an intelligible fashion and written in standard English?

Reviewer #1: Yes

Reviewer #2: Yes

5. Review Comments to the Author

Reviewer #1: The authors highlight valid challenges associated with sample collection in remote locations where researchers may not always be in a position to follow optimal transport and storage guidelines. The objective of the study is clearly outlined, the method well described, data appear appropriately analysed and conclusions are supported by the results. Some minor revisions will help to clarify certain aspects of the manuscript. Extra discussion points and recommendations for future research would lend further depth to the discussion section.

Abstract:

Line 65: Clarify if 3 or 4 horses were involved (see comments below).

Introduction:

Please provide references of the 'limited information on acceptable short-term storage conditions' that do exist.

Perhaps the authors could provide a thought on why information on this subject is limited if there is a need for more flexibility in short-term storage of samples.

Materials and Methods:

Please clarify the number of horses used in the study. Three horses are mentioned in lines 65 and 147. Four horses are mentioned in line 146 and four samples per group are listed in Table 2.

Please elaborate on how the sufficient number of samples required for statistical comparison of groups was obtained.

Line 146: Please abbreviate UCD prior to this mention or spell out the college name.

Please provide the time of year the samples were collected.

Please clarify for the reader at this point which group is the one following to manufacturer's recommendation.

Please clarify how 4 and 19 degrees were maintained and measured over 7 days.

Line 182: Please explain why a modification to the manufacturer's instructions was made.

Line 185: Please change µl to µL.

Line 187: Please change uL to µL.

Line 189: Please remove parentheses around RIN.

Line 194: Should it be 20 µL instead of 20 ul (as in line 211)? Please amend if that is the case.

Line 196: Please remove the full stop after '3.0 µL'.

Results:

Line 237-39: Please remove the speculation on why sample 8 yielded lower quantity of RNA from the results section. This needs to be discussed more thoroughly in the discussion section.

Line 241: Please remove the sentence concerning the numerically higher result in group B. The differences between groups are highlighted in the previous sentence. As there was no statistically significant difference between groups the numerically higher value of one group is irrelevant.

Table 2: Please add group names to the table and/or table description. Please include a column showing the horse associated with each sample. It was speculated that sample 8 had thicker hair which may have affected results. If the animal in question had thicker hair than its samples may have had lower results in all groups. As there are a number of samples with lower RNA quantities it would be interesting to see if they were from the same animal.

Discussion section:

Line 288: Remove the word 'to' from 'before to freezing'.

Line 312: Please elaborate on:

a) How thickness was assessed

b) How much thicker hairs in this sample were relative to those from other samples

c) How thickness may have affected results

d) Whether the 'thicker' hair from the animal that provided sample 8 also yielded lower RNA quantities in the other groups

Please discuss which cycle phase (anagen, catagen or telogen) samples may have been in at the time of sample collection. Could samples at a different phase have reacted differently to the different storage treatments?

The authors mention that limited information on this subject has been published so far. Please discuss what future research may be needed to improve the current level of available information. For example:

a) In remote locations with higher ambient temperatures than 19 degrees

b) Should this study be repeated using hair follicles at different hair cycle phases

Reviewer #2: Dear authors,

The topic of your article is certainly interesting as it could provide useful information for sampling in the field.

I also believe that the collection and handling of samples and the extraction of RNA have been done properly.

My main issue is the sample size that you have used and the subsequent conclusions that you draw from that.

With small sample size, it is quite probable to detect only large differences with reasonable power. Small differences might go undetected.

So my first question is why you decided on a sample size of 4. Could you report the power of the experiment so that it is clear that you had reasonable chance to detect differences (if they were there)?

Secondly, from table 2, I cannot infer which horse delivered which samples. Because from the text, I understand that you sample 3 bunches of hair from each of the 4 horses . Is that correct?

If the sampling was done in that way, why not using 'horse' as the second factor in a model. That brings me to the second remark of doing a one-way ANOVA. With horse and treatment, you could do a two way ANOVA which could be more powerful.

And, did you check if assumptions for performing ANOVA were fulfilled (normally distributed residuals, homogeneous variance)?

Finally, and this is more a conceptual remark, you are trying to prove equivalence of your treatment B and C to the standard method of A. This is different from trying to find a difference.

Text is written fluently and the article is concise.

L60. HF can be an easily accessible tissue but this sentence is very general. Not all gene expression in mammals can be interrogated! So this sentence should be rewritten.

L70 and following. to be reconsidered after re-analyzing.

L147 three horses or 4?

L229 I do not really agree with the analysis. See comments above.

6. PLOS authors have the option to publish the peer review history of their article (what does this mean?). If published, this will include your full peer review and any attached files.

Reviewer #1: No

Reviewer #2: No

---

## [Author Response · Author response to Decision Letter 0]

17 Jan 2024

All responses to the editor and reviewers are contained within the Cover letter which has been uploaded as the Rebuttal letter document.

---

## [Decision Letter · Decision Letter 1]

26 Feb 2024

PONE-D-23-34914R1Evaluation of short-term hair follicle storage conditions for maintenance of RNA integrity.PLOS ONE

Dear Dr. Murphy,

Thank you for submitting your manuscript to PLOS ONE. After careful consideration, we feel that it has merit but does not fully meet PLOS ONE’s publication criteria as it currently stands. Therefore, we invite you to submit a revised version of the manuscript that addresses the points raised during the review process.

We look forward to receiving your revised manuscript.

Kind regards,

Abeer El Wakil, PhD

Academic Editor

PLOS ONE

Journal Requirements:

**Additional Editor Comments:**

The concept of the present study is interesting, but one of the reviewers still has some concerns that need to be addressed.

Reviewers' comments:

Reviewer's Responses to Questions

**Comments to the Author**

1. If the authors have adequately addressed your comments raised in a previous round of review and you feel that this manuscript is now acceptable for publication, you may indicate that here to bypass the “Comments to the Author” section, enter your conflict of interest statement in the “Confidential to Editor” section, and submit your "Accept" recommendation.

Reviewer #1: All comments have been addressed

Reviewer #2: (No Response)

2. Is the manuscript technically sound, and do the data support the conclusions?

Reviewer #1: Yes

Reviewer #2: Partly

3. Has the statistical analysis been performed appropriately and rigorously? 

Reviewer #1: Yes

Reviewer #2: No

4. Have the authors made all data underlying the findings in their manuscript fully available?

Reviewer #1: Yes

Reviewer #2: Yes

5. Is the manuscript presented in an intelligible fashion and written in standard English?

Reviewer #1: Yes

Reviewer #2: Yes

6. Review Comments to the Author

Reviewer #1: I thank the authors for their detailed responses. With the edits made I believe the manuscript will be a valuable addition to the scientific literature on this topic.

Reviewer #2: Dear authors,

Thank you for reworking the article.

1) An explanation to the question of availability of the data…

Reviewer #1: Yes

Reviewer #2: No

We are unclear why Reviewer 2 has responded ‘No’ here as all of the data has been

provided in the manuscript.

I initially choose for “No” because the horse ID was not provided. This is now corrected so you have provided all data.

2) I still miss answers to the statistical issues that I raised.

a) I disagree with you on the feasablity of a two-way anova. You have a perfectly crossed design. The 3 levels of factor “storage” are observed in all 4 horses (second factor). Horse can be considered as a blocking-factor.

So you could perfectly fit a 2 way model but assuming there is no interaction between storage*horse or you could use more elaborate models.

Some refs : Applied Linear Statistical Models, Chapter 20 by Kutner et al, https://users.stat.ufl.edu/~winner/sta4211/ALSM_5Ed_Kutner.pdf or

Alin A, Kurt S. Testing non-additivity (interaction) in two-way ANOVA tables with no replication. Statistical Methods in Medical Research. 2006;15(1):63-85. doi:10.1191/0962280206sm426oa

b) On the sample size and the power:

I agree that often for transcriptomics and gene expression a very small number of samples is used. But I guess that is often also because the cost of RNAsequencing is prohibitive.

My point is that you conclude from your experiment “no significant effects between storage methods”. That is true based on your data but it could be due to small sample size and/or low power of the experiment, which would only allow for large effects to be detected.

One option is that you compute the power of the Ftest on your data retrospectively and add this in the discussion. You might have had only sufficient power (80%) to detect differences of the size of 2.5*standard error, based on, for example, TABLE B.12 Table for Determining Sample Size for Analysis of Variance (fixed factor levels model Applied Linear Statistical Models, in https://users.stat.ufl.edu/~winner/sta4211/ALSM_5Ed_Kutner.pdf)

And considering your answer…

“Small differences in RIN value or RNA concentrations for example are not meaningful when such high mean values were achieved in the results. The minimum acceptable RIN value for an RNAseq analysis from samples derived from human hair was 7 (Zhang et al., 2017). All but one sample in the current study resulted in RNA with a RIN value less than 9.4 The average total yield of RNA in the current study was 6,321 ng (mean concentration/uL x 15 uL elution volume. To put this in perspective, Kim et al., (2006) in their study evaluating gene expression in human head hair follicles, yielded total RNA of <20 ng in half of their samples”.

I do agree but why do would you do a statistical analysis (ANOVA) on the data if everything is acceptable?

c) Finally, and this is more a conceptual remark, you are trying to prove equivalence of your treatment B and C to the standard method of A. This is different from trying to find a difference.

.....You answer: "We agree, and believe the results do show equivalence of treatments to the standard

method of A."

But you did not test equivalence statistically so you can not really conclude this from the analysis.

7. PLOS authors have the option to publish the peer review history of their article (what does this mean?). If published, this will include your full peer review and any attached files.

Reviewer #1: **Yes: **Christiane O'Brien

Reviewer #2: No

---

## [Author Response · Author response to Decision Letter 1]

7 Mar 2024

Dear Reviewer,

We thank you for your additional comments which we feel we have now addressed in full as detailed below.

Reviewer #2: Dear authors,

Thank you for reworking the article.

1) An explanation to the question of availability of the data…

Reviewer #1: Yes

Reviewer #2: No

We are unclear why Reviewer 2 has responded ‘No’ here as all of the data has been

provided in the manuscript.

I initially choose for “No” because the horse ID was not provided. This is now corrected so you have provided all data.

Author response: Thank you.

2) I still miss answers to the statistical issues that I raised.

a) I disagree with you on the feasablity of a two-way anova. You have a perfectly crossed design. The 3 levels of factor “storage” are observed in all 4 horses (second factor). Horse can be considered as a blocking-factor.

So you could perfectly fit a 2 way model but assuming there is no interaction between storage*horse or you could use more elaborate models.

Some refs : Applied Linear Statistical Models, Chapter 20 by Kutner et al, https://users.stat.ufl.edu/~winner/sta4211/ALSM_5Ed_Kutner.pdf or

Alin A, Kurt S. Testing non-additivity (interaction) in two-way ANOVA tables with no replication. Statistical Methods in Medical Research. 2006;15(1):63-85. doi:10.1191/0962280206sm426oa

Author response: Respectfully, we have examined in-depth the reference provided by the reviewer and appreciate the points raised, but still do not agree that a two-way ANOVA is appropriate to use here. A two-way ANOVA is used when there are two factors that can contribute variability in the response variable. For example, had we wanted to examine the three storage treatments from samples collected from two groups of horses (e.g. sick versus healthy), or from samples collected from horses at different time points (e.g. morning versus evening), then a two-way ANOVA would be an appropriate model and an interaction between the factors could then be estimated. In the study by Alin and Kurt (2006) cited by the reviewer, two medical data sets are used as examples to explain methods to test interaction in two-way ANOVA tables with no replication. However, in one example three treatments are tested on two different types of rabbit heart (in normal and hypercholesterolemic) and in the second sample, three treatments are tested on three groups of rats. Both of these represent suitable data sets for two-way ANOVA analysis.

We strongly believe that the four horses in this study represent subjects or experimental units, and not a blocking factor. Samples were collected at the same time under the same conditions from each subject. In a completely randomised one-way ANOVA design, subjects would have been randomly assigned to each treatment and this would have required 12 horses to provide a sample size per treatment of n=4. Instead, to reduce inter-individual variability, each horse in this study contributed a sample to each treatment. Based on this, and through further evaluation of the suitability of our statistical method spurred by the reviewer’s points, we see that a repeated measures one-way ANOVA is more appropriate. This compares the mean differences between the three treatments and also accounts for inter-individual variation between subjects. We have re-run all of the statistical tests for each parameter using repeated measures one-way ANOVA. While P values have changed slightly, the overall finding of no difference between treatments for any parameter holds.

b) On the sample size and the power:

I agree that often for transcriptomics and gene expression a very small number of samples is used. But I guess that is often also because the cost of RNAsequencing is prohibitive.

My point is that you conclude from your experiment “no significant effects between storage methods”. That is true based on your data but it could be due to small sample size and/or low power of the experiment, which would only allow for large effects to be detected.

One option is that you compute the power of the Ftest on your data retrospectively and add this in the discussion. You might have had only sufficient power (80%) to detect differences of the size of 2.5*standard error, based on, for example, TABLE B.12 Table for Determining Sample Size for Analysis of Variance (fixed factor levels model Applied Linear Statistical Models, in https://users.stat.ufl.edu/~winner/sta4211/ALSM_5Ed_Kutner.pdf)

Author response: We thank the reviewer for suggesting we compute the power of the F test retrospectively. Based on a Post-hoc examination of our study findings for the RIN values, using the B12 Table from the Kutner et al, 2005, (Applied Linear Statistical Models Fifth edition) reference provided and assuming a=0.05, b=0.80, r=3 (conditions), Δ/σ= 9.7 (Max value observed in present study) – 7 (minimum reported value from literature) / 0.48 (SD) => Δ/σ= 5.6, the table indicates that our sample size of n=4 is sufficient for adequate power. We have now added to the discussion that we have retrospectively confirmed by using the cited reference table that the sample size for the study was adequate. We have also addedthe Kutner et al reference to the bibliography.

And considering your answer

“Small differences in RIN value or RNA concentrations for example are not meaningful when such high mean values were achieved in the results. The minimum acceptable RIN value for an RNAseq analysis from samples derived from human hair was 7 (Zhang et al., 2017). All but one sample in the current study resulted in RNA with a RIN value less than 9.4 The average total yield of RNA in the current study was 6,321 ng (mean concentration/uL x 15 uL elution volume. To put this in perspective, Kim et al., (2006) in their study evaluating gene expression in human head hair follicles, yielded total RNA of <ng in half of their samples”.

I do agree but why do would you do a statistical analysis (ANOVA) on the data if everything is acceptable?

Author response: Point noted, thank you.

c) Finally, and this is more a conceptual remark, you are trying to prove equivalence of your treatment B and C to the standard method of A. This is different from trying to find a difference.

.....You answer: "We agree, and believe the results do show equivalence of treatments to the standard

method of A."

But you did not test equivalence statistically so you can not really conclude this from the analysis.

Author response: We agree, that was poor wording in our response. Failure to find a difference is not the same as establishing equivalence.

---

## [Decision Letter · Decision Letter 2]

19 Apr 2024

PONE-D-23-34914R2Evaluation of short-term hair follicle storage conditions for maintenance of RNA integrity.PLOS ONE

Dear Dr. Murphy,

Thank you for submitting your manuscript to PLOS ONE. After careful consideration, we feel that it has merit but does not fully meet PLOS ONE’s publication criteria as it currently stands. Therefore, we invite you to submit a revised version of the manuscript that addresses the points raised during the review process.

We look forward to receiving your revised manuscript.

Kind regards,

Abeer El Wakil, PhD

Academic Editor

PLOS ONE

Journal Requirements:

Reviewers' comments:

Reviewer's Responses to Questions

**Comments to the Author**

1. If the authors have adequately addressed your comments raised in a previous round of review and you feel that this manuscript is now acceptable for publication, you may indicate that here to bypass the “Comments to the Author” section, enter your conflict of interest statement in the “Confidential to Editor” section, and submit your "Accept" recommendation.

Reviewer #2: (No Response)

2. Is the manuscript technically sound, and do the data support the conclusions?

Reviewer #2: Yes

3. Has the statistical analysis been performed appropriately and rigorously? 

Reviewer #2: Yes

4. Have the authors made all data underlying the findings in their manuscript fully available?

Reviewer #2: Yes

5. Is the manuscript presented in an intelligible fashion and written in standard English?

Reviewer #2: Yes

6. Review Comments to the Author

Reviewer #2: Dear authors,

Thank you for the submitted review.

I can agree with your choice to do a repeated, one way ANOVA to test for the effect of storage. By doing so you consider the within-subject correlation in your data. But I believe my suggestion of using a two-factor model would also work.

With respect to the power I do not agree with the computations or the wording on line 330. When you do a post-hoc analysis you need to use the averages of your data. In that case the computation of delta is not appropriate. Delta is the max(mean)-min(mean) in the data. For RIN your means are 9.475, 9.25 and 9.725 so the delta is 0.475 and the RootMSE=0.48 which leads to a ratio of approx..= 1 . The power to detect differences of this size is only 17%.

Δ/σ= 9.7 (Max value observed in present study) – 7 (minimum reported

value from literature) / 0.48 (SD) => Δ/σ= 5.6, the

However if you would argue that you wanted to test if RIN values would be too low for any treatment (e.g. RIN=7) then I could follow your reasoning but that is not a proper post hoc test. So you could rephrase that you wanted to detect very large differences for which n=4 is sufficient.

7. PLOS authors have the option to publish the peer review history of their article (what does this mean?). If published, this will include your full peer review and any attached files.

Reviewer #2: No

---

## [Author Response · Author response to Decision Letter 2]

7 May 2024

Reviewer #2: Dear authors,

Thank you for the submitted review.

I can agree with your choice to do a repeated, one way ANOVA to test for the effect of storage. By doing so you consider the within-subject correlation in your data. But I believe my suggestion of using a two-factor model would also work.

Authors:We are glad that you are in agreement with our choice, thank you.

With respect to the power I do not agree with the computations or the wording on line 330. When you do a post-hoc analysis you need to use the averages of your data. In that case the computation of delta is not appropriate. Delta is the max(mean)-min(mean) in the data. For RIN your means are 9.475, 9.25 and 9.725 so the delta is 0.475 and the RootMSE=0.48 which leads to a ratio of approx..= 1 . The power to detect differences of this size is only 17%.

Δ/σ= 9.7 (Max value observed in present study) – 7 (minimum reported

value from literature) / 0.48 (SD) => Δ/σ= 5.6, the

However if you would argue that you wanted to test if RIN values would be too low for any treatment (e.g. RIN=7) then I could follow your reasoning but that is not a proper post hoc test. So you could rephrase that you wanted to detect very large differences for which n=4 is sufficient.

Authors: Thank you for this clarification. We have expanded this section and rephrased the wording on line 330 to state that we were interested in detecting large differences only, for which n=4 was considered sufficient. We have supported this by highlighting samples sizes used from similar previously published papers and referred to the differences between our results and the minimum accepted RIN value from the literature. We have removed the Kutner reference again.

---

## [Editor Report · Decision Letter 3]

15 May 2024

Evaluation of short-term hair follicle storage conditions for maintenance of RNA integrity.

PONE-D-23-34914R3

Dear Dr. Murphy,

We’re pleased to inform you that your manuscript has been judged scientifically suitable for publication and will be formally accepted for publication once it meets all outstanding technical requirements.

Kind regards,

Abeer El Wakil, PhD

Academic Editor

PLOS ONE
---

## [Editor Report · Acceptance letter]

20 May 2024

PONE-D-23-34914R3 

PLOS ONE

Dear Dr. Murphy, 

I'm pleased to inform you that your manuscript has been deemed suitable for publication in PLOS ONE. Congratulations! Your manuscript is now being handed over to our production team.

Kind regards, 

on behalf of

Professor Abeer El Wakil 

Academic Editor

PLOS ONE